# Embodied Semantic Scene Graph Generation

**Xinghang Li, Di Guo, Huaping Liu**∗**, Fuchun Sun**
Department of Computer Science and Technology, Tsinghua University, China
Beijing National Research Center for Information Science and Technology, China
∗Corresponding author: Huaping Liu (hpliu@tsinghua.edu.cn)

**Abstract:** Semantic scene graph provides an effective way for intelligent agents to better understand the environment and it has been extensively used in many robotic applications. Existing work mainly focuses on generating the scene graph from the sensory information collected from a pre-defined path, while the environment should be exhaustively explored with a carefully designed path in order to obtain a comprehensive semantic scene graph efficiently. In this paper, we propose a new task of Embodied Semantic Scene Graph Generation, which exploits the embodiment of the intelligent agent to autonomously generate an appropriate path to explore the environment for scene graph generation. To this end, a learning framework with the paradigms of imitation learning and reinforcement learning is proposed to help the agent generate proper actions to explore the environment and the scene graph is incrementally constructed. The proposed method is evaluated on the AI2Thor environment using both the quantitative and qualitative performance indexes. Additionally, we implement the proposed method on a streaming video captioning task and promising experimental results are achieved.

**Keywords:** Semantic Scene Graph, Embodied Exploration, Learning for Visual Navigation

## 1 Introduction

The scene graph is a collection of nodes in a graph structure where nodes usually represent scene entities and edges represent geometrical transformation between the nodes. It provides a flexible way to track objects and their spatial relations within the scene. Recently, the concept of scene graph is successfully adapted in many applications in computer vision and robotics [1][2]. For example, the Visual Genome dataset [3], which contains annotations of objects and their relationships in images, has been extensively utilized in various scene graph generation tasks [4][5][6]. In [1], the author proposes to extract the knowledge graph from recorded videos for better video understanding. Xu et al. [2] provides an extensive survey on the recent progress on the generation and applications of the scene graph.

Furthermore, 3D semantic scene graph is of great interest for robotic applications such as navigation, mapping, and interaction [7]. To

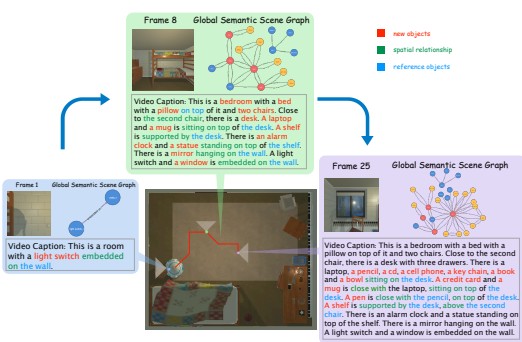

Figure 1: The illustration for the Embodied Semantic Scene Graph Generation. The agent moves around and always selects an appropriate action to obtain better semantic scene graph. Further, the generated semantic scene graph can be helpful to generate the streaming video captioning.

understand the 3D environment, Kim et al. [8] proposes a 3D scene graph construction framework using RGB-D data processing pipeline, which can be used for task planning and visual question answering. Bozcan and Kalkan [9] develops a Boltzmann Machines-based generative scene model bringing objects, their spatial relations and affordances together. Armeni et al. [10] presents a 3D

5th Conference on Robot Learning (CoRL 2021), London, UK.

scene graph, which unifies multi-modal semantic information in the 3D space. Recently, Rosinol et al. [11] proposes a 3D dynamic scene graph which extends to represent dynamic scenes with moving agents.

Though great success has been achieved for semantic scene graph, all of the above work focuses on generating scene graph from the sensory information collected from a pre-defined path. However, to obtain a comprehensive and informative semantic scene graph, the environment should be exhaustively explored, which requires large time consumption. In many scenarios, the disconnection between the scene graph generation and the exploration path even results in unsatisfied results. Therefore, it is necessary for the agent to have the ability to autonomously explore the environment to generate a good semantic scene graph.

In this work, we adopt the embodied exploration framework to tackle the *Embodied Semantic Scene Graph Generation* problem, which has never been addressed before. It combines the action and perception abilities of the agent. The embodied agent generates actions to autonomously explore the environment and then incrementally constructs a high-quality semantic scene graph using the detected object instances and contextual information. The collected sensor data is continuously processed and incorporated into the scene graph model along with the exploration process. The illustration is demonstrated in Fig.1. Different from the existing work which focuses on generating scene graphs from acquired sensor information, our work introduces the embodiment of the agent to effectively combine the action, vision and language together for autonomous 3D semantic scene graph generation. The main contributions are summarized as follows:

1. We propose a new framework for the Embodied Semantic Scene Graph Generation problem, which exploits the embodiment characteristic of the intelligent agent to find an appropriate path for scene graph generation in an embodied environment.

2. We develop a learning framework with paradigms of imitation learning and reinforcement learning to help the agent acquire the intelligence to generate high-quality scene graph.

3. We test the proposed method on the AI2Thor dataset and evaluate its effectiveness using the quantitative and qualitative performance indexes.

For the rest of the paper, Section II presents the related work. Problem formulation and architecture are introduced in Section III. Sections IV and V give details about the scene graph generation and navigation modules. Section VI presents the experimental results and Section VII closes this paper.

## 2 Related Work

This work falls into the intersection of the embodied exploration and semantic understanding. Therefore, we present a brief review on both domains.

The embodied exploration lies in the intersection of the computer vision and robotics. The representative work includes active sensing [12][13], embodied question answering [14], embodied captioning [15], embodied amodal detection [16], multi-agent embodied exploration [17], etc. In these works, one or multiple embodied agents equipped with visual perception modules can autonomously explore the environment to perform various active sensing and perception tasks. Zhang and Mei [18] proposes a constructive model for collective intelligence in which each agent would explore and communicate to cooperate with each other. In recent years, researchers have combined deep reinforcement learning with active vision sensing. Han et al. [19] builds an active vision system based on DQN to guide the agent generate appropriate actions and get better images for detection. Furthermore, Chaplot et al. [20] studies the task of embodied interactive learning for object detection. It should be noticed that in [15], the authors propose an embodied captioning task, which is relevant to our work, while our work solves a more fundamental task of generating the 3D scene graph, which provides more comprehensive contextual information of the environment. Actually, the generated scene graph can be useful for many down-streaming tasks such as image captioning, streaming video captioning, embodied Questions and Answering, etc.

On the other hand, the embodied exploration task can also benefit from the semantic scene graph. Druon et al. [21] and Zeng et al. [22] study how to enhance the active object search using the spatial context and semantic linking maps respectively. The hierarchical mechanical search also shows performance improvement by using the semantic modeling [23]. Du et al. [24] proposes to learn

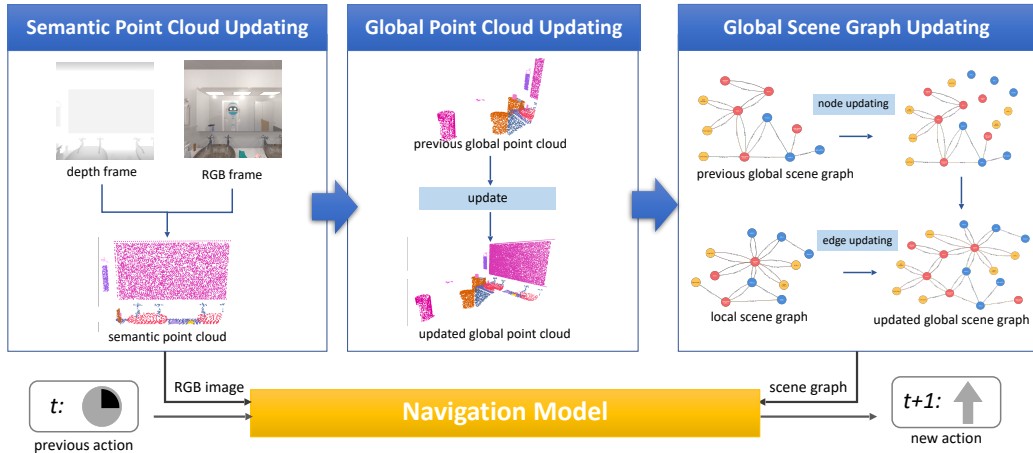

Figure 2: The architecture of the *Embodied Semantic Scene Graph Generation*. At time instant $t$, after taking one action, the agent comes to a new viewpoint and updates the local and global scene graph, and further exploits the local and global scene graph to generate a new action. This process iterates until a *stop* action is obtained.

object relation graph and tentative policy for visual navigation. Besides, the scene graph also plays an important role in the visual question answering system for robot manipulation [25]. These works illustrate the significance of semantic scene graph in embodied perception. However, they all use existing scene graphs and do not address the generation of the scene graph.

## 3 Problem Formulation and Architecture

The goal of this work is to enable the agent to automatically generate a sequence of actions to explore the environment and build the corresponding semantic scene graph incrementally. Concretely speaking, the agent starts from any location of a scene with an initial empty global semantic scene graph $\mathcal{GSSG}_0$. At time instant $t$, the agent could construct a local scene graph $\mathcal{LSSG}_t = \{\mathcal{N}_t^{(l)}, \mathcal{R}_t^{(l)}\}$ from its observation $s_t$, which could be extracted from observed RGB and depth images. The local scene graph $\mathcal{LSSG}_t$ should be merged with the previous global scene graph $\mathcal{GSSG}_{t-1} = \{\mathcal{N}_{t-1}^{(g)}, \mathcal{R}_{t-1}^{(g)}\}$ to get the updated global scene graph $\mathcal{GSSG}_t = \{\mathcal{N}_t^{(g)}, \mathcal{R}_t^{(g)}\}$. Please note that we follow the conventional symbols with slight modifications [26], i.e. $\mathcal{N}_t^{(l)}$ and $\mathcal{R}_t^{(l)}$ represent the node and semantic relationships for the local scene graph $\mathcal{LSSG}_t$, and $\mathcal{N}_t^{(g)}$ and $\mathcal{R}_t^{(g)}$ represent the node and semantic relationships for the global scene graph $\mathcal{GSSG}_t$. After that, a navigation module should produce an action $a_t$ for the agent to obtain the next observation $s_{t+1}$. The above procedure iterates until a satisfactory global semantic scene graph is achieved and the stop action is triggered. The obtained semantic scene graph can then be used to describe the environment comprehensively and solve some downstream tasks such as captioning, question answering, etc.

We call this problem as *Embodied Semantic Scene Graph Generation*, which exploits the embodiment capability of the agent to collect data and construct high-quality scene graph. It is significantly different from existing scene graph generation work such as [8][10]. To solve this problem, we design an architecture as shown in Fig.2. It is mainly composed of the scene graph generation and navigation module. For the scene graph generation, a 3D semantic point cloud is produced from the RGB and depth images and is used for generating the local scene graph, which is then used to incrementally update the global scene graph. For the navigation module, we utilize the RGB frame, previous action and the scene graph as input and an action is generated for the next step.

# 4 Scene Graph Generation

The scene graph generation module can be divided into two parts: local scene graph prediction and global scene graph generation, which are introduced as follows.

**Local scene graph prediction:** we mainly follow the graph convolution network based scene graph generation method which is able to pass the information between objects. It is widely used in tasks that utilize the scene graph to bridge the gap between semantic and visual information [26][27]. The generated local semantic scene graph includes the current in-sight objects and their relations with each other. We additionally introduce object class embedding and bounding box coordinates as the input for each object node, and the training objective is to restore the bounding box coordinates and labels for each object and edge. Please see the supplementary material for more details.

**Global scene graph generation:** At each time step, the agent takes an action to move, and the equipped camera captures the RGB and depth frames of the scene to construct a local scene graph, which is further merged into the global scene graph. During this period, the detected objects in the local scene graph and the existing objects in the previous global scene graph are aligned.

Different from the *multi-view consistency* considered in [10], which has the 3D mesh and multi-view visual information known in advance, we do not have any prior information about the scene and the information of each object is incrementally updated. Considering this situation, we propose a point cloud based weighted voting mechanism which utilizes the number of the pixels in the point cloud to measure the confidence of the object category. In practice, for each detected object, we maintain a score distribution across predefined classes. The score is incrementally accumulated by the weighted value of the confidence provided by the detector and the size of the acquired point cloud.

Then, we align the object point cloud with the nodes in global scene graph by computing the fraction of point cloud in the object point cloud that is inside the node's 3D bounding box. We calculate this fraction between each object point cloud and the node in the global scene graph. If the highest fraction of object point cloud of the corresponding node is higher than the preset threshold and this fraction is higher than those with other objects, the object and the node are considered aligned. This bidirectional alignment procedure could ensure that each aligned object and the corresponding node are in one-to-one correspondence. Fig.3 demonstrates the alignment process. Based on the aligning result for each object and the

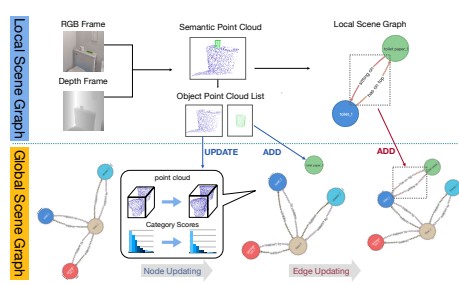

Figure 3: The alignments between objects and edges of local scene graph and global scene graph.

weighted voting mechanism, we could align the nodes and edges in the obtained Local Semantic Scene Graph(LSSG) with those in the Global Semantic Scene Graph(GSSG). The updating operation could involve ADD, UPDATE and REPLACE for both nodes and edges in GSSG. Please see the supplementary material for more details.

# 5 Navigation Model

## 5.1 Model Structure

The goal of the proposed navigation model is to guide the agent to take actions to explore the environment and build the semantic scene graph incrementally. The action space considered in this work includes *Move*, where the agent can take nine basic actions corresponding to 8 directions and no move, and *Rotate*, where the agent can rotate clockwise or counterclockwise with a fixed angle $\delta_r = 90°$ or no rotate. Especially, if no move and no rotate are selected, a stop action is triggered.

The design of the navigation model is a challenging problem since the agent is expected to build a good semantic scene graph which contains the objects and their relationships accurately. To tackle this problem, we construct a navigation architecture as shown in Fig.4. Considering that the $LSSG$ could represent the insight semantic observations for the agent, we use an LSTM to embed the

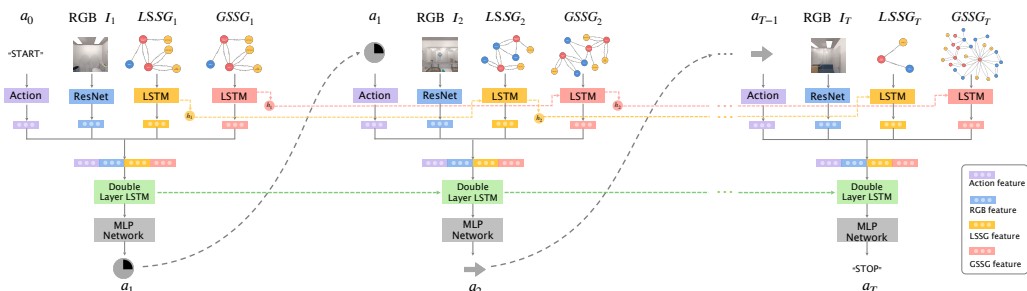

Figure 4: The architecture of the navigation model. At each time step, the of action feature, RGB feature, LSSG feature and GSSG feature are firstly extracted and concatenated and fed into a double-layer LSTM to predict the next step action.

$LSSG$ sequence to help the agent build connections over visited viewpoints on a semantic level. Further, we introduce $GSSG$ vector aiming at help the agent have a complete perception over the global semantic information. Since the change of $GSSG$ could reflect the adding and updating of objects and semantic relations in the scene, we utilize an LSTM to make the agent capture this dynamic process. Overall, we use a ResNet to extract the visual feature, two LSTMs to encode the local scene graph and global scene graph sequence respectively, and an embedding module to represent the previous action. The three feature vectors are concatenated and fed into a double-layer LSTM model. Finally, an MLP network is used to generate the next step action.

## 5.2 Learning for Navigation

Since the direct learning for such an architecture is still challenging, we adopt the imitation learning as the pre-training and the reinforcement learning for fine-tuning.

### 5.2.1 Imitation Learning

The goal of imitation learning for sequential prediction problem is to train the agent to mimic the expert's behaviors. In this work, it is important to generate some demonstration paths for the agent to imitate. We adopt a two-stage method to deal with this problem. In the first stage, we try to obtain a way-point set and in the second stage, we perform the interpolation between way-points to get the whole demonstration path. To evaluate the way-points, we first count the visible object at each viewpoint, and select the closest next way-point that has the most number of unseen new objects, which can be expressed as:

$$v^* = \underset{v \in O(v_c, k^*)}{\arg\max}\ new\_object\_num(v),\ k^* = \underset{k}{\arg\min} \sum_{v \in O(v_c, k)} new\_object\_num(v) > 0$$

where $O(v_c, k)$ represents the feasible viewpoints that are $k$ steps away from $v_c$, and the way-point set can be updated as $\mathcal{W} = \mathcal{W} \cup \{v^*\}$. The above procedure is repeated until a maximum distance is achieved. In the second step, we implement a beam search over the way-point sequence for interpolation [28]. The loss function is defined as follows:

$$\mathcal{L}_\theta = -\frac{1}{K} \sum_{k=1}^{K} \sum_{t=1}^{T_k} \log \pi_\theta(\hat{a}_{k,t} | \hat{s}_{k,0}, \hat{a}_{k,0}, \hat{s}_{k,1}, \hat{a}_{k,1}, \cdots, \hat{s}_{k,t}) \tag{1}$$

where $K$ is the number of demonstration paths used for training in one batch, $T_k$ is the length of the $k$-th path, $\hat{s}_{k,t}$ and $\hat{a}_{k,t}$ are the annotated input visual state and action, and $\theta$ denotes the parameter of the exploration policy $\pi$. The process of minimizing $\mathcal{L}_\theta$ equals to maximize the probability of the demonstration paths' action sequence based on the annotated inputs.

### 5.2.2 Reinforcement Learning

After pre-training the exploration model with imitation learning, we then try to further improve its performance using the REINFORCE algorithm.

Since our task aims at formulating a scene graph for the entire scene, we directly measure the similarity between the generated scene graph and ground truth one, which can be obtained from the ground truth information about the object and the relationships. Given a global scene graph $\mathcal{GSSG}$ which is constructed from one path, the similarity score can then be calculated as a weighted sum of the precision and recall rate of the nodes and edges:

$$Sim(\mathcal{GSSG}) = \lambda_{node}(R_{node} + \lambda_p P_{node}) + R_{edge} + \lambda_p P_{edge} \tag{2}$$

where $P_{node}$ and $P_{edge}$ are the precision of the nodes and edges, respectively, while $R_{node}$ and $R_{edge}$ are the recall rate of the nodes and edges, respectively.

On the other hand, we hope to encourage the observation diversity when constructing the scene graph. The diversity can be characterized as the number of observation viewpoints of the detected objects. Concretely speaking, given a global scene graph $\mathcal{GSSG}$ which is constructed from one path, we use $O$ to denote the set of the detected objects and calculate the diversity as:

$$Div(\mathcal{GSSG}) = \sum_{o \in O} num\_viewpoints(o) \tag{3}$$

where $num\_viewpoints(o)$ is the number of the viewpoints about the detected object instance $o$. Therefore we may formulate the score at time instant $t$ as:

$$p_t = Sim(\mathcal{GSSG}) + \lambda_d Div(\mathcal{GSSG}) - \rho t. \tag{4}$$

where the third term is used to penally the length of the path, and $\lambda_d$, $\rho$ are the corresponding weighting parameters. In practice, we set $\lambda_{node} = 0.1$, $\lambda_p = 0.5$, and $\lambda_d = \rho = 0.001$.

According to the above definition, the immediate reward is designed to be the increment of the score $r(s_t, a_t) = p_t - p_{t-1}$ and the cumulative reward can be computed as:

$$R(s_t, a_t) = r(s_t, a_t) + \sum_{t'=t+1}^{T} \gamma^{t'-t} r(s_{t'}, a_{t'}), \tag{5}$$

where $R(s_t, a_t)$ represents the expected accumulated reward when agent takes action $a_t$ at state $s_t$, the discount parameter $\gamma$ is set to 0.99, and $T$ is the length of the action and state sequence with upper bound of 40 steps, and we use SGD optimizer with a learning rate of $10^{-4}$.

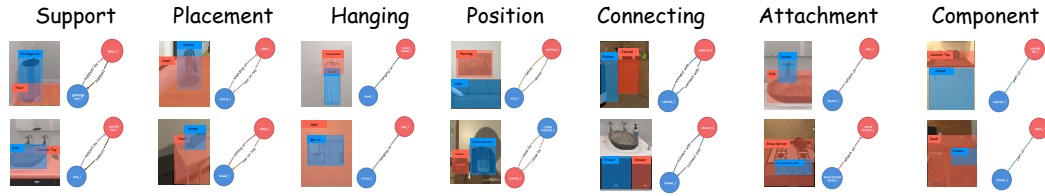

Figure 5: An illustration of the considered semantic relationships.

## 6 Experimental Validation

### 6.1 Dataset

We choose AI2THOR [29] to generate our dataset since it supports a continuous navigation while MP3D[30] does not, and its has higher rendering quality of images and more accurate sensors compared with Habitat[31]. We follow the guidelines in [26] to extract 16 semantic relationships, which can be clustered into the following seven categories (Fig.5). Please refer to the supplementary material for more details.

Table 1: Performance Comparison

| | RelCls | | | | | | | SGGen | | | | | | |
|---|---|---|---|---|---|---|---|---|---|---|---|---|---|---|
| | NoS | $P_{node}$ | $R_{node}$ | $F1_{node}$ | $P_{edge}$ | $R_{edge}$ | $F1_{edge}$ | NoS | $P_{node}$ | $R_{node}$ | $F1_{node}$ | $P_{edge}$ | $R_{edge}$ | $F1_{edge}$ |
| Random | 27.367 | 1.0 | 0.311 | 0.474 | 0.705 | 0.144 | 0.239 | 27.367 | 0.550 | 0.223 | 0.317 | **0.278** | 0.097 | 0.144 |
| PPO+Map | 23.915 | 1.0 | 0.484 | 0.652 | **0.736** | 0.260 | 0.384 | 23.915 | **0.665** | 0.435 | 0.526 | 0.248 | 0.200 | 0.221 |
| Frontier-based | 24.597 | 1.0 | 0.547 | 0.707 | 0.717 | 0.289 | 0.412 | 24.597 | 0.643 | 0.425 | 0.512 | 0.253 | 0.194 | 0.220 |
| LSTM+Act+Pose | 22.815 | 1.0 | 0.563 | 0.720 | 0.689 | 0.300 | 0.418 | 22.815 | 0.567 | 0.526 | 0.546 | 0.235 | 0.208 | 0.221 |
| LSSG+GSSG(Ours) | 22.435 | 1.0 | **0.688** | **0.815** | 0.681 | **0.339** | **0.453** | 26.657 | 0.558 | **0.564** | **0.561** | 0.228 | **0.230** | **0.229** |
| Traverse(Oracle) | >500 | 1.0 | 0.858 | 0.924 | 0.576 | 0.384 | 0.461 | >500 | 0.516 | 0.617 | 0.562 | 0.201 | 0.275 | 0.232 |

Table 2: Input Ablation Study: Baseline takes RGB and last step action as input, LSSG and GSSG represent local and global semantic scene graph vector respectively.

| | RelCls | | | | | | | SGGen | | | | | | |
|---|---|---|---|---|---|---|---|---|---|---|---|---|---|---|
| | NoS | $P_{node}$ | $R_{node}$ | $F1_{node}$ | $P_{edge}$ | $R_{edge}$ | $F1_{edge}$ | NoS | $P_{node}$ | $R_{node}$ | $F1_{node}$ | $P_{edge}$ | $R_{edge}$ | $F1_{edge}$ |
| Baseline | 22.815 | 1.0 | 0.563 | 0.720 | 0.689 | 0.300 | 0.418 | 22.815 | 0.567 | 0.526 | 0.546 | **0.235** | 0.208 | 0.221 |
| GSSG | 17.254 | 1.0 | 0.577 | 0.732 | **0.694** | 0.309 | 0.428 | 22.942 | 0.524 | 0.460 | 0.490 | 0.228 | 0.200 | 0.213 |
| LSSG | 23.778 | 1.0 | 0.657 | 0.793 | 0.678 | 0.337 | 0.450 | 22.829 | 0.541 | 0.539 | 0.540 | 0.229 | 0.213 | 0.224 |
| LSSG+GSSG | 22.435 | 1.0 | **0.688** | **0.815** | 0.681 | **0.339** | **0.453** | 26.657 | 0.558 | **0.564** | **0.561** | 0.228 | **0.230** | **0.229** |

Table 3: Training Paradigm Ablation Study: IL represents imitation learning, RL represents reinforcement learning, IL+RL refers to our final training paradigm.

| | RelCls | | | | | | | SGGen | | | | | | |
|---|---|---|---|---|---|---|---|---|---|---|---|---|---|---|
| | NoS | $P_{node}$ | $R_{node}$ | $F1_{node}$ | $P_{edge}$ | $R_{edge}$ | $F1_{edge}$ | NoS | $P_{node}$ | $R_{node}$ | $F1_{node}$ | $P_{edge}$ | $R_{edge}$ | $F1_{edge}$ |
| IL | 32.477 | 1.0 | 0.632 | 0.774 | 0.663 | 0.323 | 0.434 | 25.731 | 0.550 | 0.556 | 0.553 | 0.229 | 0.214 | 0.221 |
| RL | 20.423 | 1.0 | 0.573 | 0.729 | **0.694** | 0.301 | 0.420 | 15.020 | **0.619** | 0.475 | 0.538 | **0.231** | 0.197 | 0.222 |
| IL+RL | 22.435 | 1.0 | **0.688** | **0.815** | 0.681 | **0.339** | **0.453** | 26.657 | 0.558 | **0.564** | **0.561** | 0.228 | **0.230** | **0.229** |

## 6.2 Method and Metrics

To verify the effectiveness of our framework and the influence of the input vectors, we introduce the following methods for comparison:

1. **Random** The agent randomly selects an action from action space to perform scene exploration.

2. **PPO+Map[32]** The agent utilizes Resnet-18 to encode RGB frame, local map and global map respectively and concatenates them as the input for the LSTM to predict the next action.

3. **Frontier-based Exploration[33]** A heuristic algorithm that guides the agent to detect and sweep to the closest frontier with path planning.

4. **LSTM+Act+Pose[14, 15]** This model is widely used in embodied tasks, it encodes RGB frame, agent position and last step action as the input of LSTM to predict the next step action for the agent.

5. **LSSG+GSSG(Ours)** Our framework that takes LSSG and GSSG vectors as the LSTM input.

6. **Traverse(Oracle)** Ask the agent to traverse every viewpoint in the scene.

To evaluate the quality of the semantic scene graph, we adopt the precision, recall rate and F1 score of the nodes and edges compared to the ground truth scene graph, denoted as $P_{node}$, $R_{node}$, $F1_{node}$, $P_{edge}$, $R_{edge}$ and $F1_{edge}$ respectively, and the number of steps $NoS$ as quantitative metrics. Note that $R_{node}$ could be viewed as an object level coverage, and $R_{edge}$ could represent the semantic richness of the generated $\mathcal{GSSG}$.

We also propose the following two tasks for evaluation. **RelCls:** In this task, we align the detecting result from Mask RCNN with ground truth to obtain the correctly detected objects at each viewpoint, and offer the ground truth segmentation of each object to the agent. Since the detection results and point cloud information are always correct, the precision of node is always 1.0. **SGGen:** In this task, the agent utilizes the raw detecting result from Mask RCNN which is consistent with real-world environment. The agent is expected to leverage both the global scene graph and navigation policy to deal with the imperfection of Mask RCNN and the incompleteness of the collected point cloud, incrementally generating an accurate and informative scene graph.

## 6.3 Results

### 6.3.1 Quantitative results

The experimental results are shown in Table I and II, from which we have the following observations:

1. In both RelCls and SGGen, $LSSG+GSSG$ outperforms other baselines on the recall rate and F1-score of both the node and edge, and achieves a performance close to Traverse, which demonstrates the effectiveness of our framework. Note that Traverse asks the agent to traverse every viewpoint in the unseen scene which always requires manual assistance in reality. While our method fully exploits the embodied capacity of the robot to explore the scene automatically and reaches a comparable performance with much fewer $NoS$.

2. The increase in recall rate is often accompanied by a decrease in precision. This is mainly caused by the imperfection of the Mask RCNN and Local Scene Graph Prediction Network(LSGPN) since the errors at some specific viewpoints could be hard to update and would accumulate as the agent increasingly covers the scene. Our method achieves the highest F1-score on both tasks, which illustrates that our framework handles the balance of precision and recall rate well.

3. SGGen is much more difficult than RelCls since both the label and the mask of the detecting result could be wrong, and the detecting error could further affect the generated $LSSG$ and $GSSG$. This explains the drop of all metrics in SGGen compared with those in RelCls. No matter how difficult the task is, the baseline methods only implement spatial exploration and adopt the same exploring path. While our method explores the scene at a semantic level indicating that the agent is aware of $GSSG$ and would try to update and add the semantic information of $GSSG$ in subsequent steps.

4. From table II we could discover that in RelCls, both $LSSG$ and $GSSG$ benefit the task, and the improvement of introducing $LSSG$ is more obvious. This may be because the $LSSG$ sequence encoded by LSTM contains a certain amount of $GSSG$ information. While in SGGen, adding $GSSG$ vectors is worse than Base method. We think this could be caused by the more frequently changing frequency on $GSSG$ and its vector is very different from that in the training. $LSSG + GSSG$ performs best on both tasks, which illustrate the effectiveness and robustness of our framework.

5. Table III shows the performance of our $LSSG + GSSG$ method under different training paradigm. We can see that IL performs better than RL in RelCls. In SGGen, the performance has an obvious drop as the behaviour cloning based method utilized by IL has poor generalization ability. In unseen testing scene, especially the ones in SGGen that have huge differences in aspect of room layout and point cloud accuracy with training scenes, IL trained model tends to perform much worse. Meanwhile, while RL trained model has a smaller performance drop in SGGen, it has the lowest recall rate over nodes and edges in RelCls and SGGen, which demonstrates that the scene has not been completely explored. Therefore, we design the mix training paradigm that utilizes imitation learning to first pre-train the model, and then finetunes it with reinforcement learning. Our method (IL+RL) achieves the highest score over the recall rate and f1 score of nodes and edges on both two tasks, demonstrating its efficiency.

### 6.3.2 Qualitative Analysis and Applications

The generated semantic scene graph can be used for various downstream tasks. And we have applied the generated semantic scene graph in a streaming video captioning task in this paper. The caption of the video is incrementally enriched with the generated scene graph while the agent exploring the environment. Experiment results show that the quality of the generated caption is superior to that generated with a $Random$ exploration policy. The detailed results and further discussion are presented in the supplementary document. Please also visit our website https://embodiedscenegraph.vercel.app/ for more information.

## 7 Conclusion

In this paper, a novel embodied semantic scene graph generation framework is established and the hybrid imitation reinforcement learning method is developed to address this new task. With the learned policy, the agent is able to autonomously explore the environment and then incrementally generate a high-quality semantic scene graph. This research exploits the important embodiment characteristic of the intelligent agent and paves a new path for the agent to semantically describe the environment. Additionally, it provides a fundamental component for some challenging downstream tasks such as streaming video captioning, robotic manipulation, visual navigation, etc. We would also like to extend this work to large-scale scene graph generation in more complicated scenarios.

## Acknowledgements

This work was supported in part by the National Natural Science Fund for Distinguished Young Scholars (62025304) and in part by the Seed Fund of Tsinghua University (Department of Computer Science and Technology)-Siemens Ltd., China Joint Research Center for Industrial Intelligence and Internet of Things.

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
