# OpenReview forum: "Embodied Semantic Scene Graph Generation"
_robot-learning.org/CoRL/2021/Conference — CoRL2021 Poster_

### Official Review · Reviewer_vave · 2021-07-23

**Originality:** Good
**Technical Quality:** Good
**Clarity Of Presentation:** Good
**Impact:** 3

**Recommendation:**

Weak Reject: I recommend rejecting the paper, but will not argue for my recommendation if the majority of other reviewers have a different opinion.

**Summary:**

The paper studies a new visual navigation problem named “Embodied Semantic Scene Graph Generation”. The task is to have an embodied agent (e.g., a navigation robot) equipped with a first-person view camera traversing an indoor environment and generating a global semantic scene graph. A semantic scene graph consists of object categories and their semantic relationships. The goal is to generate a scene graph that closely resembles a hidden ground truth scene graph of the environment. The paper presents a method that learns to carry out this task by imitating traversal demonstrations and reinforcement learning. The paper also shows that the generated scene graphs are useful for a downstream video captioning task.

**Issues:**

See main comments.

**Reviewer Expertise:**

Very good: Comprehensive knowledge of the area

**Strengths And Weaknesses:**

I like the overall problem setup of embodied semantic scene graph generation. Semantic scene graph is a useful representation for reasoning about the high-level structure of an environment, which can be useful as a guidance downstream tasks like multi-step manipulation and object searching. The problem of simultaneously navigating the environment and incrementally generating a global scene graph that matches a hidden ground truth graph is a good starting point for more future works along this line. I also appreciate the effort of carrying out a large-scale study in the AI2-THOR environment. I hope the authors release the corresponding dataset in the future.

At the same time, I’m not entirely convinced that the paper is sufficient as the first study of the problem. My main concern is that the paper doesn’t really have any discussion on why the problem of embodied scene graph generation is challenging. Intuitively, there are at least two challenges behind the problem: 1) how to *efficiently* traverse the space to generate the complete scene graph and 2) generalizing the policy to a new environment.

The introduction section briefly touches upon (1) by saying that (line 51) “the environment should be 51 exhaustively explored, which requires large time consumption”. However, the proposed method squarely bypasses the problem by providing demonstrations generated by a heuristics-based policy as training data. And the key contribution of the method, i.e., to incorporate the previously collected scene graph information in decision making, isn’t designed to address this problem.

For (2), I’m not entirely sure about the problem setup. Are there training and held-out scenes or are all scenes used for training? I don’t see it mentioned anywhere in the main text of the supplementary material. I hope the author spends more text discussing the key challenges of this problem as well as the potential solutions for them.

Relatedly, I’m not entirely clear on the motivation behind the navigation module. The only explanation is at Line 167-169: “The design of the navigation model is a challenging problem since the agent is expected to build a good semantic scene graph which contains the objects and their relationships accurately. To tackle this problem, we construct a navigation architecture which is shown in Fig.4.” This isn’t really a justification of why it might help the agent build a better semantic scene graph, nor does it explain why it might outperform other baselines. I hope the author provides a more thorough discussion on the method itself and how it might address the key challenges presented in the problem setup.

Minor: Wrong citation at line 125. The goal of work [23] is to generate images from scene graph, not the other way around.

**Summary Of Recommendation:**

Like mentioned in my comment, I like the new problem setup. But I'm not entirely sure that the paper did an adequate job addressing the key challenges of this problem setup and pointing at directions for more follow up works. In particular, the paper did not justify some of the key components of the proposed method and why it might outperform other comparable baselines. Thus I recommend rejecting before seeing more comments from the authors.

---

> ### Author Response · Authors · 2021-08-26
> **reply to reviewer vave**
>
> **[W1] The introduction section briefly touches upon (1) by saying that (line 51) “the environment should be 51 exhaustively explored, which requires large time consumption”. However, the proposed method squarely bypasses the problem by providing demonstrations generated by a heuristics-based policy as training data. And the key contribution of the method, i.e., to incorporate the previously collected scene graph information in decision making, isn’t designed to address this problem.**
>
> For problem (1), we design a heuristic algorithm to generate demonstrating trajectories in order to exhaustively explore the environment with an appropriate path. The demonstrating trajectories aim at maximizing the number of objects and the viewing angles of each object. Firstly, we utilize imitation learning to make the agent clone the behavior of the heuristic algorithm and obtain a fine performed model. And then, we design a reward that takes the trajectory length, precision and recall rate of the nodes and edges in global semantic scene graph into account and the reinforcement learning is used to further finetune the model. After the training, the trained model is supposed to generate appropriate action instructions for the agent to exhaustively explore the environment and generate a satisfying semantic scene graph. It can be seen from table 1 that with the trained model, we obtain comparable results with far less number of steps when comparing to the oracle method, which traverses every viewpoint in the scene. It demonstrates that the proposed model is able to generate appropriate exploration path to nicely explore the environment and generate good semantic scene graph.
>
> **[W2] For (2), I’m not entirely sure about the problem setup. Are there training and held-out scenes or are all scenes used for training? I don’t see it mentioned anywhere in the main text of the supplementary material. I hope the author spends more text discussing the key challenges of this problem as well as the potential solutions for them.**
>
> Actually, four scene categories are considered in the proposed task. For each scene category, there are 30 rooms, among which we utilize 26 rooms for training, 2 rooms for evaluation and the left 2 rooms for testing. Therefore, the result shown in the paper are the performance of the agent in unseen scenes demonstrating the generalization ability of the proposed method. Meanwhile, our mixing training paradigm of imitation learning and reinforcement learning aims at solving the problem (2). With only imitation learning, it lacks the generalization ability and the performance significantly decreases in unseen scenes. Therefore, we introduce the reinforcement learning to finetune the model to improve the generalizability capability in unseen scenes. To further illustrate this problem, we have added the ablation study on training paradigm, namely the IL method, RL method, and IL+RL method. Generally, the IL+RL method yields the best performance. And it is clear to see that the IL+RL method is superior to that of IL method, which reflects the effectiveness of the RL in finetuning the model in the proposed task. For more details, please see section 6 and table 3 in the revised paper.
>
> **[W3] Relatedly, I’m not entirely clear on the motivation behind the navigation module. The only explanation is at Line 167-169. This isn’t really a justification of why it might help the agent build a better semantic scene graph, nor does it explain why it might outperform other baselines. I hope the author provides a more thorough discussion on the method itself and how it might address the key challenges presented in the problem setup.**
>
> We have reorganized the paper and added more illustration of the navigation model and detailed explanation of the training paradigm in section 5.1 and section 5.2 respectively. Specifically, at each time step, the of action feature, RGB feature, LSSG feature and GSSG feature are firstly extracted and fed into a double-layer LSTM to predict the next step action, where the LSSG could represent the insight semantic observation, and the GSSG could help the agent to have a complete perception over the global semantic information. As the GSSG changes while exploring, we utilize an LSTM to make the agent capture these dynamic processes. To further analyze the navigation model, we have implemented an ablation study across different inputs (only RGB, LSSG, GSSG, LSSG+GSSG) for the navigation architecture and the results are illustrated in table 2. Meanwhile, you could also watch our video on the anonymous website https://embodiedscenegraph.vercel.app for a more intuitive demonstration.
>
> **[W4] Minor issue: wrong citation at line 125. The goal of work [23] is to generate images from scene graph, not the other way around.**
>
> We have fixed this wrong citation. Additionally, we have performed a proofreading throughout the paper.

---

### Official Review · Reviewer_bACY · 2021-07-23

**Originality:** Very Good
**Technical Quality:** Very Good
**Clarity Of Presentation:** Very Good
**Impact:** 4

**Recommendation:**

Weak Accept: I recommend accepting the paper, but will not argue for my recommendation if the majority of other reviewers have a different opinion.

**Summary:**

This work presents an approach for learning to explore in order to build a semantic scene graph. The approach leverages prior work on semantic scene graph construction (which nominally uses precollected exploration data) and expands upon this to enable exploration. The exploration is learned from initially imitation learning based on a scripted policy and then reinforcement learning to improve the policy. Further details of graph construction in this setting are described. The approach is demonstrated on an AI2Thor environment to perform well.

**Issues:**

The performance of the approach outperforms previous approaches, though only by a small margin (particularly for SGGen), but is nonetheless is interesting. There are a few open questions I believe that are left unanswered and should be discussed:
- From the learning side, how much does the RL contribute to exploring well
- Ablations across inputs to the navigation architecture would be useful (e.g., removing the RGB, LSSG, and GSSG)
- The oracle compared traverses every node, it would be useful to see how the proposed method compares to an oracle at a similar number of steps.
- Add std deviations (potentially in the appendix to values)

The paper also lacks qualitative downstream performance of the scene graph. Though the qualitative analysis of scene graph construction is reasonable, the primarily goal of the scene graph is to perform well on downstream tasks like captioning or planning. It would be interesting to see how the performance on such tasks is qualitatively changed compared to baselines.

Minor notes:
- Text on figure 1, 3, 4 is really small
- Typo L70 “3. We testify” -> we test
- Add more information in the captions of table 1 and 2 on what each value means.


**Reviewer Expertise:**

Good: General knowledge of the area

**Strengths And Weaknesses:**

Strengths:
- The paper is well written and clear
- The setting of embodied exploration to build a representation is interesting
- The approach is interesting and reasonable. The description of how the scene graph is built from local observations into global observations with voting is useful.
- The network architecture for navigation which captures the local and global scene graph is well done.

Weaknesses:
- The comparisons are reasonable, though more qualitative downstream performance would be useful (e.g., qualitatively test on planning or captioning)
- The figures of scene graphs are difficult to see and understand due to their size, though the figures are generally good
- A few ablations on the training and architecture would be useful


**Summary Of Recommendation:**

Overall the paper is interesting and well written. The approach makes sense and is valuable. There are a few issues of clarity of the figures and tables that can be easily addressed. There are also some additional ablations and experiments that would help to reinforce the results.

---

> ### Author Response · Authors · 2021-08-26
> **reply to reviewer bACY**
>
> **[W1] From the learning side, how much does the RL contribute to exploring well.**
>
> We have added the ablation study on training paradigm, namely the IL method, RL method, and IL+RL method.  Unsurprisingly, the IL+RL method yields the best performance. And it is clear to see that the IL+RL method is superior to that of IL method, which reflects the effectiveness of the RL in the proposed task. For more details, please see section 6 and table 3 in the paper.
>
> **[W2] Ablations across inputs to the navigation architecture would be useful (e.g., removing the RGB, LSSG, and GSSG).**
>
> We have already implemented an ablation study across different inputs (only RGB, LSSG, GSSG, LSSG+GSSG) to the navigation architecture and the results are illustrated in table 2. For the RGB data, as we feel that the RGB information is necessary for representing the environment in the proposed task, we do not implement the ablation of RGB data.
>
> **[W3] The oracle compared traverses every node, it would be useful to see how the proposed method compares to an oracle at a similar number of steps.**
>
> In our work, the trained navigation module will automatically generate the "stop" action when a satisfactory semantic scene graph has been achieved. Therefore, the number of steps of the proposed model is actually determined by the navigation model. In this setting, we could see that the f1 scores of the nodes and edges are close those with oracle. If we force the agent to keep navigating after it chooses to stop, the agent might revisit explored viewpoints and the generated scene graph cannot well reflect the performance of the proposed model.
>
> **[W4] The paper also lacks qualitative downstream performance of the scene graph.**
>
> We have added a qualitative comparison experiment between our method and the random method for the video captioning task in the supplementary document. We also believe that it is meaningful to further investigate the downstream applications of the semantic scene graph in the future.
>
> **[W5] Add std deviations (potentially in the appendix to values)**
>
> For the test experiment, the agent starts from the same starting view point, and our navigation model is deterministic. Therefore, the trajectories of the agent and the corresponding metrics would not change. Considering this situation, we don’t add std deviations.
>
> **[W6] Minor issues**
>
> + We have enlarged the font size in Fig.1, Fig.3 and Fig.4 and adjusted the layout of the figures to make them look clearer.
> + More explanations for table 1 and table 2 are added. Additionally, we have performed a proofreading throughout the paper.

---

### Official Review · Reviewer_iKq5 · 2021-07-26

**Originality:** Excellent
**Technical Quality:** Excellent
**Clarity Of Presentation:** Excellent
**Impact:** 4

**Recommendation:**

Strong Accept: I recommend accepting the paper and will argue for my recommendation even if other reviewers hold a different opinion.

**Summary:**

The paper considers the problem of embodied scene graph generation, in which, the agent sequentially constructs a scene graph by interacting with the environment. At every time step, it chooses an action that determines the next observation, generates a local scene graph from the observation, and then integrates the local scene graph with the global scene graph constructed thus far.

The action sequence is modeled by an LSTM network that takes in input features from the observation (RGB-D scan) and LSTM on the two – local and global – constructed scene graphs. The action sequence is pre-trained with imitation learning to pick a waypoint that maximizes the number of unseen objects. The model is then trained using RL with a cost function that is a combination of precision and recall.


**Issues:**

1. Section 6.4 seems rather much. It would help the paper if instead of presenting this application – which does not go in with the main theme of the paper – could present some more details on the architecture that have been relegated to the appendix section; especially 5.1 which describes the model structure in just one-two line.

2. The conclusion seems short for such a nice paper.

3. I could not find a description of how miss detections are dealt with. For instance, if the agent constructs a node in the global scene graph, by a misdetection, but realizes that the object wasn’t what it thought it was or that there was no object there in future scans. How does the agent correct it?

Minor:
1. References with missing year: [15].
2. In 5.2.1, explain what is O(v,k)?
3. Section 1 mentions “this work embodiment of the agent to effectively combine the action, vision, and language together for autonomous 3D semantic scene graph generation” Where is the language part in the proposed architecture?
4. Figures 3 and 4 are too small to read


**Reviewer Expertise:**

Excellent: Expert knowledge on the topic of the paper

**Strengths And Weaknesses:**

Strengths: The paper introduces a problem of immense significance and does solid work.

Weaknesses: The paper attempts to convey too many things and finds eight pages limiting.

**Summary Of Recommendation:**

The paper studies a very pertinent problem of embodied scene graph generation. The proposed solution is well investigated. The paper is nicely written. Overall, the paper is a good, solid work that I have enjoyed reading.

The only limitation would be that the paper tries to do too much. Section 6.4 seems rather too much and could be left out as a future work or in the appendix. The space could have been used to explain the model architecture and the RL reward function more clearly in the main paper.

---

> ### Author Response · Authors · 2021-08-26
> **reply to reviewer iKq5**
>
> **[W1] Section 6.4 seems rather much. It would help the paper if instead of presenting this application – which does not go in with the main theme of the paper – could present some more details on the architecture that have been relegated to the appendix section; especially 5.1 which describes the model structure in just one-two line.**
>
> As suggested, we have reorganized our paper accordingly. For Section 6.4, we have moved it to the supplementary document. And we have added more illustration of the proposed architecture and detailed explanation of the training paradigm in Sections 5.1-5.2 respectively
>
> **[W2] The conclusion seems short for such a nice paper.**
>
> We have extended the conclusion by emphasizing the novelties of the proposed task, and further applications and discussions for the future work.
>
> **[W3] I could not find a description of how miss detections are dealt with. For instance, if the agent constructs a node in the global scene graph, by a misdetection, but realizes that the object wasn’t what it thought it was or that there was no object there in future scans. How does the agent correct it?**
>
> We do consider the situation of missing detection, while due to the page limit, the detailed illustration is presented in the supplementary document (page 3 and 4 in section 2: Global Scene Graph Generation). Concretely speaking, we propose a weighted voting mechanism to alleviate the influences of imperfect detection. The weighted voting mechanism maintains the score distribution over all categories for each object. For each step, we get the semantic point cloud of the view. Then we would calculate the intersection between the point cloud of each detected object and the 3D bounding box of each node in the global semantic scene graph to align the detected object with the graph node. For the aligned graph node, we update the score of a specific category by adding the product of detector confidence and point number proportion (normalized with the resolution of the RGB observation) of the detected object. If the category with the highest score changes, we would update the category of the node and set the semantic point cloud as the new point cloud of the node. Otherwise，we would merge the point cloud of the detected object with the point cloud of the node. After aligning and updating the node, the edges in local semantic scene graph and global semantic scene graph with same subject and object node would be naturally aligned. We maintain the highest confidence for each edge in global semantic scene graph, and would update the confidence and category of each edge in global semantic scene graph with the corresponding edge in local semantic scene graph. With this mechanism, the missing or wrong detection could be modified as the agent explores the environment. Fig. 3 is an illustration of the node aligning, score updating and edge aligning process.
>
> **[W4] Minor issues**
>
> + We have complete the publication year of Ref.[15].
> + O(v,k) represents the set of viewpoints that are k steps away from viewpoint v.
> + Actually, the language part is more related to the generated semantic scene graph which contains rich semantic information representing as language. And the semantic scene graph can be used to improve the performance many language relevant downstream tasks such as streaming video caption. Therefore, we consider that our work combines the action, vision and language together for autonomous 3D semantic scene graph generation.
> + We have enlarged the font size both in Fig.3 and Fig.4 and adjusted the layout of the figures to make them look clearer. Additionally, we have added the caption for Fig.4 to further illustrate the figure.

---

### Official Review · Reviewer_nnax · 2021-07-27

**Originality:** Fair
**Technical Quality:** Very Good
**Clarity Of Presentation:** Fair
**Impact:** 3

**Recommendation:**

Weak Accept: I recommend accepting the paper, but will not argue for my recommendation if the majority of other reviewers have a different opinion.

**Summary:**

The paper proposes a new embodied AI task "Embodied semantic scene graph generation" in which the objective is for an agent to select actions to efficiently build a semantic scene graph. It also proposes an approach to solve this problem based in imiation and reinforcement learning.

**Issues:**

 - lack of clarity of exposition
 - lack of detail in methodology
 - lack of clear novelty

**Reviewer Expertise:**

Good: General knowledge of the area

**Strengths And Weaknesses:**

Strenths:

The generation of scene graphs is an interesting and important problem in embodied navigation.

Weaknesses:

[W1] The related work should also cover the literature related to active sensing which is very related to the problem at hand.

[W2] The approach only seems to be evaluated on relatively small scenes that are on the order of 1 room and take on the order of 20 steps to explore. It would be interesting to see how the method scales to larger environments.

[W3] The work seems to be a collection of components (local scene graph generation, local scene graph fusion, action selection, imitation learning, reinforcement learning) none of which on their own strikes me as particularly novel. The core claim of the paper is that the novelty is in the action selection part, but this component of the paper is relatively under-explained with many details left out and seeming to be a relatively standard.

Smaller comments:
 - The text in Figure 1 is very small and hard to read.
 - replace "Ref. [X]" with "Authorname et. al."
 - The manuscript should be proofread for grammar and spelling mistakes of which there are many
 - We can't see from Figure 6 what actions the agent took.


**Summary Of Recommendation:**

While I would agree that the embodied scene graph generation task is an interesting one, I find that the paper is not very clear and lacks significant novelty. Rather, it comes across more as a collection of components that have been combined. This does not rule it out as systems work is still important, but I don't feel that the bar has been achieved that the work is of significant enough importance to warrant publication at CoRL.

---

> ### Author Response · Authors · 2021-08-26
> **reply to reviewer nnax**
>
> **[W1] The related work should also cover the literature related to active sensing which is very related to the problem at hand.**
>
> We agree that active sensing is very related to the proposed problem and it is actually one of the foundations of our embodied exploration. We have included the relevant ones in the paper. Please check line 80-87, Section 2 in our revised submission.
>
> **[W2] The approach only seems to be evaluated on relatively small scenes that are on the order of 1 room and take on the order of 20 steps to explore. It would be interesting to see how the method scales to larger environments.**
>
> Actually, we consider all the four scene categories, which leads to 30*4=120 rooms. For each scene category, we use 26 rooms for training, 2 rooms for evaluation and the remaining 2 rooms for testing. Since this work mainly aims at solving the newly proposed task of Embodied Scene Graph Generation, we provide intuitive results in the widely-used AI2THOR environment. Additionally, we have noticed that there are some other large scale datasets such as MP3D and Habitat released recently. However, MP3D does not support continuous navigation and therefore we cannot use it for validations. It seems that Habitat simulator could provide a new testbed for our work and it merits further investigation. For this point, we give a brief discussion in the revised version (please see line 228-230 in section 6.1)
>
> **[W3] The work seems to be a collection of components (local scene graph generation, local scene graph fusion, action selection, imitation learning, reinforcement learning) none of which on their own strikes me as particularly novel. The core claim of the paper is that the novelty is in the action selection part, but this component of the paper is relatively under-explained with many details left out and seeming to be a relatively standard.**
>
> We would like to claim that the main contribution of our paper is to propose a novel embodied scene graph generation task which exploits the embodiment of the intelligent agent to autonomously explore the environment and generate a semantic scene graph. In this setting, the action and perception of the agent need to be seamlessly combined to achieve an embodied exploration task. To solve this novel problem, we propose a new framework which is composed of the scene graph generation and navigation module. For the scene graph generation, the local semantic scene graph in every step is used to incrementally update the global semantic scene graph. And for the navigation module, the visual information, previous generated local and global semantic scene graph, and previous action are fed into the navigation module to generate the exploration action for the next step. To promote the performance of the framework, we develop a training paradigm of imitation learning and reinforcement learning to help the agent generate proper actions to explore the environment. We have included details of our framework and training paradigm in the supplementary document. We also claim that many components used in this framework could be replaced by more recently proposed ones to achieve better performance.
>
> **[W4] Minor issues**
> + We have enlarged Fig.1 to make it clearer and the reference format is corrected as suggested.
> + For Fig.6, we have added the action information and moved it to the supplementary document for further clarification.
> + Additionally, we have performed a proofreading throughout the paper.

---

> > ### Comment · Reviewer_nnax · 2021-09-02
> > **Response to rebuttal**
> >
> > The authors have done a reasonable job of addressing my concerns. I am happy to raise my rating to weak accept.

---

### Meta-Review · Area_Chair_mcFP · 2021-08-10

**Recommendation:** Accept (Poster)
**Confidence:** 4

**Metareview:**

This paper proposed a new task of embodied semantic scene graph generation, where the goal is to have an embodied agent intelligently select the actions to traverse an environment and build a semantic scene graph of the environment incrementally from its own experiences. At the initial reviews, this paper received mixed ratings, with two Weak Rejects, one Weak Accept, and one Strong Accept. While the authors appreciated the new problem formulation, concerns have been raised in model novelty, writing clarity, and experimental validation. The meta-reviewer strongly suggested the authors respond to the issues suggested by the reviewers. For the reviewers, please take a look at each other's reviews and the authors' responses. Let's try our best to resolve the conflicting ratings and reach a consensus.

**Update:** The authors revised their manuscript and addressed the issues brought up by the initial reviews. At the end of the discussion period, Reviewer nnax updated their rating to Weak Accept and the other three reviewers retrained their original scores. Reviewer vave, who voted Weak Reject, argued that some of their concerns had been addressed by the authors' responses, yet this paper needs substantial revision to convince the reader that: 1) the new problem is interesting and challenging, and 2) the proposed solution is a meaningful starting point to address this problem. The other three reviewers appraised the problem formulation of embodied scene graph generation. They believed that the proposed model makes sense as a first attempt to tackle this problem, even though individual components of this model did not present significant novelty. The AC understood the rationale behind the positive and negative comments of this work. Considering all the factors, the AC believed that such a paper that explored this new topic should be accepted, given the fact that three of the four reviewers have supported its acceptance and no critical flaw has been spotted during the review process. The authors should further improve the quality of this manuscript by incorporating all the reviewers' comments.

---

> ### Author Response · Authors · 2021-08-31
> **reply to area chair mcFP**
>
> Thank you for the feedback and insightful comments! A modified version of our paper together with the supplementary document is updated in the system. In the following dialogue boxes, we have responded to all reviewers' comments in details respectively.
>
> Generally, we have reorganized our paper and more technical details about the navigation model and training paradigm are added in the main paper. Additionally, we have included more ablation studies on the training paradigms to further verify the effectiveness of the proposed method. For more qualitative experimental results, we have included them in the supplementary document. We believe that this modified version of the paper will better clarify the contributions and technical details of our paper. We hope these modifications may help improve reviewers‘ confidence in our paper. Thank you!

---

### Decision · Program_Chairs · 2021-09-13

**Decision:**

Accept (Poster)

**Comment:**

This paper proposed a new task of embodied semantic scene graph generation, where the goal is to have an embodied agent intelligently select the actions to traverse an environment and build a semantic scene graph of the environment incrementally from its own experiences. At the initial reviews, this paper received mixed ratings, with two Weak Rejects, one Weak Accept, and one Strong Accept. While the authors appreciated the new problem formulation, concerns have been raised in model novelty, writing clarity, and experimental validation. The meta-reviewer strongly suggested the authors respond to the issues suggested by the reviewers. For the reviewers, please take a look at each other's reviews and the authors' responses. Let's try our best to resolve the conflicting ratings and reach a consensus.

**Update:** The authors revised their manuscript and addressed the issues brought up by the initial reviews. At the end of the discussion period, Reviewer nnax updated their rating to Weak Accept and the other three reviewers retrained their original scores. Reviewer vave, who voted Weak Reject, argued that some of their concerns had been addressed by the authors' responses, yet this paper needs substantial revision to convince the reader that: 1) the new problem is interesting and challenging, and 2) the proposed solution is a meaningful starting point to address this problem. The other three reviewers appraised the problem formulation of embodied scene graph generation. They believed that the proposed model makes sense as a first attempt to tackle this problem, even though individual components of this model did not present significant novelty. The AC understood the rationale behind the positive and negative comments of this work. Considering all the factors, the AC believed that such a paper that explored this new topic should be accepted, given the fact that three of the four reviewers have supported its acceptance and no critical flaw has been spotted during the review process. The authors should further improve the quality of this manuscript by incorporating all the reviewers' comments.